# Decree-Law 54/2018: Perspectives of Early Childhood Educators on Inclusion in Preschool Education in Portugal

Rita Simas Bonança [1], Paulo César Fróes Bulhões [2], Levi Leonido [3,4,*] and Elsa Maria Gabriel Morgado [5]

1   International Ibero-American University, Campeche 24560, Mexico; rita.bonanca@gmail.com
2   Faculty of Social Sciences and Humanities, University of the Azores, 9500-321 Ponta Delgada, Portugal; paulo_bulhoes82@hotmail.com
3   School of Human and Social Sciences, University of Trás-os-Montes e Alto Douro, 5000-801 Vila Real, Portugal
4   Center for Research in Arts Sciences and Technologies, Portuguese Catholic University, 4169-005 Porto, Portugal
5   Center in Basic Education, Instituto Politécnico de Bragança, 5300-253 Bragança, Portugal; elsa.morgado@ipb.pt
*   Correspondence: levileon@utad.pt

**Abstract:** In this study, we analyze the perspectives of early childhood educators working in various teaching sectors on inclusion in preschool education, following the publication of Decree-Law no. 54/2018, as of 6 July, at a time when learning and inclusion support measures are being extended to all children and the SARS-CoV-19 (COVID-19) pandemic prevails in Portugal and worldwide. Based on a mixed methodological approach and the application of a questionnaire survey of 250 early childhood educators, we reflect on the implementation of the legal framework, involvement of the multidisciplinary learning and inclusion support team (EMAEI), teacher training, pedagogical/collaborative work, and mobilized support for preschool education children. The results obtained indicate quality and effectiveness in the pedagogical plan, and collaborative work between teachers and specialist technicians, although higher education in the field of inclusive education does not seem to provide professionals with the necessary and in-depth knowledge on the subject. They also indicate that, alongside dissatisfaction with the insufficient support provided to preschool education children, the relevant diploma is not fully applied, with doubts still remaining about its application.

**Keywords:** education; preschool education; inclusion; public policies



## 1. Introduction

In Portugal, the development of early childhood education (ECE) emerged after the April 25th Revolution in 1974. As for guiding principles for childhood, they were only defined for the first time in 1997 in the Curricular Guidelines for Early Childhood Education (OCEPE) by the Ministry of Education (ME), and revised only in 2016, nearly two decades later. During this period, the pedagogical framework of ECE (1997) largely focused on the principles of developmental psychology. However, in the new version of the OCEPE (2016), aspects of curricular nature were clarified with greater rigor as a result of experienced difficulties and following contributions that emerged from a national debate, which supported all the work developed [1]. In this context, Ferreira and Tomás [2] (p. 80). argue that ECE has undergone greater institutionalization, advocating that in the last 30 years: "Has been drawing a trajectory in which the accumulation of disciplinary knowledge and content, centered on a didactic, standardized, and uniform type of individualized transmission, seems to aim at the acquisition of formal learning and school competencies. [ . . . ] This involves promoting increasingly early literacy, numeracy, technology, scientism, and multilingualism, exercised through the intensive use of school-type manuals and/or proposals for activities focused on the transmission of school content".

According to the ECE framework Decree-Law no. 5/97, implemented on 10 February, ECE corresponds to the first stage of basic education in the lifelong education process: "being complementary to the educational action of the family, with which it must establish close cooperation, favoring the formation and balanced development of the child, with a view to their full integration into society as an autonomous, free and supportive being" [3] (article 2, p. 670).

Preschool education "is intended for children aged between 3 years and the age of entry into basic education and is taught in preschool education establishments" [3] (article 3, p. 670). Although the legislation covers children from 3 years of age, not including the Crèche, the National Council of Education (CNE) reiterates that attendance is a right of all children. With the approval of Decree-Law (DL) no. 54/2018, as of 6 July [4], the changes introduced by Law no. 116/2019 [4] on 13 September in Continental Portugal, and with the publication of Regional Legislative Decree no. 11/2020/M [5] on 29 July, which adapts the regimes set out in DL no. 54/2018 [6] and DL no. 55/2018 [7] to the Autonomous Region of Madeira, what is intended by inclusive education has been globally defined. In the autonomous region of the Azores, Regional Legislative Decree no. 17/2015/A [8], 22 June, amends Regional Legislative Decree no. 15/2006/A of 7 April, which establishes the legal regime for special education and educational support that aims to "create conditions for the adaptation of the educational process to the requirements of children and young people with special educational needs or with learning difficulties" [9] (p. 4359).

Additionally, at the legal level, the RGAPA approves the Regulation of Administrative and Pedagogical Management of Students (order no. 75/2014, 18 November) [10], including the creation and operation of educational support programs in chapter VIII and the special educational regime in chapter X. It should be noted that, currently, in the Azores, a pedagogical innovation pilot experience is being developed within the scope of inclusive education, as per order no. 1811/2018 of 12 October. It is also worth noting that a legislative proposal has already been presented for public discussion, aimed at organizing the regional education system, during the period from 1 February to 1 March 2022. With the publication of DL no. 54/2018, it can be inferred that the recent diploma also includes the universe of children enrolled in EPE, and it can be read that "this DL applies to school clusters and ungrouped schools, vocational schools and private, cooperative and solidarity preschool and basic and secondary education establishments, hereinafter referred to as schools" [6] (article 1, point 3, p. 2918).

The pandemic situation (March 2020) coincides with the approval of the legal regime that regulates inclusive education in Portugal. The learning of children in EPE is systematically conditioned by other contexts of stimulation. "Participation and Involvement of Families" emphasizes that "there is no study that does not confirm it: quality preschool education is one of the best predictors of future school success" [11] (p. 5). Hence, we consider it vital for research to obtain more precise data on the experienced reality in Portuguese schools about the phenomenon under study, considering the constraints caused by the spread of the SARS-CoV-2 virus. As well as understanding the impact and application of public policies on the subject under study, the main purpose of this study includes the search for a better understanding of respondents' opinions about the application of DL no. 54/2018 [6], involvement of EMAEI, quality of teacher training and pedagogical/collaborative work, and the support mobilized for children in EPE.

The national legislation in question emanates from guidelines, conceptual bases, and various assumptions agreed between member states [12–15]. There are several studies and documents guiding and evaluating their application (e.g., UNESCO; UNICEF; OECD; European Agency for Special Needs and Inclusive Education; European Commission's Directorate-General for Structural Reform Support—DG REFORM) [16,17], with the purpose of marking and guiding national legislation with supranational guidelines. Thus, the current research aims to contribute to the particular and national reflection with a global perspective in the European and international context in the framework of organizations with responsibility in evaluating and proposing changes regarding legislation and educa-

tional policies that relate (directly or indirectly) to special education and inclusion in early childhood education.

## 2. Inclusion in Preschool Education in a Pandemic Context

Inclusive education is based on the principle of equity, allowing for the success of all children and students in accessing the curriculum and essential learning, promoting what is recommended in the "Profile of Students at the End of Compulsory Education" in each and every one of the [6]. In this regard, Correia [18] argues that inclusive education is conceptually based on pedagogical freedom and community sense, as well as collaboration and justice. Thus, it reinforces that pedagogical quality is fundamental in the inclusive process, seeking to address children's needs through differentiation, considering the observed heterogeneity.

The law regulating inclusive education, approved in 2018, has given a broader scope to the concept of special educational needs, as DL no. 3/2008 of 7 January [19] only included children and young people with permanent special educational needs, based on an evaluation supported by the International Classification of Functioning, Disability and Health of the World Health Organization. DL no. 54/2018 states that inclusive education is a "process that aims to respond to the diversity of students' needs by increasing the participation of all in learning and school community life" [6] (p. 2919). It advocates for the abandonment of student categorization systems, namely, the category of special educational needs. Legislation aimed solely at special needs students guarantees a continuum of responses for all students, emphasizes educational responses rather than student categories, and ensures the mobilization, whenever necessary and appropriate, of resources from health, employment, vocational training, and social security [20].

The aforementioned legislation states that medical opinion is now optional, and that the EMAEI can make decisions on the mobilization of measures to support learning and inclusion (MSAI) at any point in children's and students' academic paths, according to their educational needs [6]. The current legal framework aims to strengthen the participation of everyone in the students' learning process. Cecílio et al. [21] and Correia [18] assert that inclusion means to involve everyone, enabling a broader and more effective education, where all children have a place, are welcomed, stimulated, and valued. Based on this principle, Castro [22], Vieira and Omote [23]), Cecílio et al. [21], Correia [18], and Esper et al. [24] ensure that inclusion does not occur in isolation, meaning that it requires the participation of both families and professionals and the community to respond to the children's conditions, ensuring new educational opportunities. Bonança et al. [25] (p. 8) emphasize that "looking at inclusion, therefore, implies understanding the concept of variability, mediating organizational transformations, in time and space and methodologies and materials, depending on the difficulties felt according to each student's profile".

Defending a similar position, the study developed by Bulhões and Condessa [26] emphasizes the role of education professionals in mediating and protecting children in order to develop favorable conditions for their learning.

Regarding the constraints caused by COVID-19, the Global Education Monitoring Report points out that exclusion during this period is limited to "Not only people with disabilities, but also others due to gender, age, place of residence, poverty, type of disability, ethnicity, indigeneity, language, religion, migration or displacement status, sexual orientation or gender identity expression, incarceration, beliefs and attitudes. It is the system and context that do not consider the diversity and multiplicity of needs, which was also highlighted by the COVID-19 pandemic" [27] (pp. 9–10).

Boer and Assino [28], Correia [18], and Esper et al. [24] argue that during the pandemic, there was a huge effort on the part of the school community; however, accessibility to digital resources for less autonomous children contributes to the increase in inequalities in learning. The pandemic has infringed upon the right to education of millions of children, and therefore, the central issue of this problem should not be defined around the recovery of pandemic effects, but on changing public policies of various countries [18,29]. In response



to this conjecture, the CNE published a set of recommendations to reduce the impact of the pandemic, arguing that "Although these recommendations are aimed at the school as a whole, their relevance is even greater the earlier the age of schooling and the more precocious the ages worked with, starting from nursery and preschool and with a strong focus on basic education" [30] (p. 3).

Giving voice to this concern, a recent study published by UNICEF reports that 40 million preschool-aged children worldwide did not have access to education due to the COVID-19 pandemic, which caused the sudden closure of nurseries and schools [31].

Regarding the diploma that regulates inclusive education, many, including early childhood educators, raised doubts due to "the fact that the measures provided in this Diploma are subject to multiple readings and forms of implementation, depending on interpretations" [32] (p. 2), generating a great heterogeneity of processes and putting into question the application of the decree, as well as the principles of equity and inclusion, enshrined in its text. Correia [18] also did not spare criticism of the new model of inclusive education, stating that the ambiguities presented in this diploma act as a brake on the success of children with special educational needs. DL 54/2018 of 6 July, which is nothing more than a decree of which the rhetoric not only intends to sell us a concept of inclusion (total inclusion) that has nothing to do with its scientific genesis and interpretation, but also takes the opportunity to extinguish concepts such as special education, special educational needs, and special needs, among others, simply to pursue the "fashion" that total inclusion brings additional benefits for all students, including those with SEN, or even to satisfy personal interests. That is, we are faced with two types of beliefs. One, objective, which fits into scientific truth, empirically confirmed, and the other, subjective, which relies on a wordy discourse borrowing from science what it cannot ensure" [18].

The criticisms and contradictions previously highlighted greatly justify the objectives and obvious need to know the degree of implementation and respective implications in context on the phenomenon under study in more detail.

*Public Network of the Ministry of Education and Training of Childhood Educators*

The recent results released in the "Inclusive Education Questionnaire 2020/2021" translate and support the tumultuous reality of exclusion that also extends to EPE children enrolled in Portuguese public schools, due to the reduced number of children covered by selective and/or additional measures and the low number of teachers and technicians to support children and students who need greater academic support, at a time when we are still suffering from the impact of the pandemic crisis. According to the conclusions of this study, there were 78,268 EPE children and students enrolled in public schools in the ME network, for whom selective and/or additional measures for learning and inclusion support were mobilized. Of the total number of children and students referenced, only 3474 EPE children were covered by selective and/or additional measures.

Let us focus on the prevalence rates of selective and/or additional measures, with a particular focus on EPE children enrolled in public schools. From reading the data, we can see that 2.1% of children benefited from selective measures, 0.1% solely benefited from additional measures, and 0.5% were benefiting from both selective and additional measures [33].

Regarding teachers who perform specific functions of learning and inclusion support, by recruitment group, we found that there are only 6611 affiliated with recruitment group 910; 242 belong to various recruitment groups; 157 belong to group 920; and 112 belong to recruitment group 930, totaling 7122 teachers in schools supporting children and students in their learning across all Portuguese schools. This means that on average, each teacher performing functions of learning and inclusion support has 11 children and/or students.

As for specialized technicians working in schools, we were able to determine that the prevalence lies with psychologists (634.9), speech therapists (366.0), and occupational therapists (137.4), totaling 1138.3 specialized technicians in schools. The number of specialized technicians working in Portuguese schools totals a value of 1508.9, a number far below what

is expected, given the number of children and students identified with selective and/or additional measures for learning and inclusion support. This means that on average, there is one specialized support technician for 51.9 children and/or students. In addition to this reality surrounding differential care for EPE children, we have identified inconsistencies in the mobilization of measures educational measures established in the referred diploma. DL no. 54/2018 [6] tells us the following about the extension of educational measures to children and students:

The measures to support learning and inclusion aim to adapt to the needs and potential of each student and ensure the conditions for their full realization, promoting equity and equal opportunities in access to the curriculum, attendance, and progression throughout compulsory education [6] (article 6, paragraph 1, p. 2921).

However, following an FAQ [20] issued by the Directorate-General for Education, we find that MSAI is limited to younger children in EPE, namely the mobilization of selective and/or additional measures, as follows: "are all measures of Decree-Law No. 54/2018, of 6 July, appropriate for Pre-School Education? No. Pre-school education is the educational level in which the curriculum is developed with full articulation of learning, in which spaces are managed flexibly, in which children are called to actively participate in planning their learning, and in which project method and other active methodologies are routinely used. The inclusion of all and each of the children in preschool education is naturally carried out through the adoption of differentiated pedagogical practices that respond to individual needs and characteristics, and it is the educator's competence to plan and design educational action based on a holistic reading of the evidence collected. Considering the above, selective and additional measures are not suitable for EPE, and all possibilities that a universal and preventive approach offers should be exhausted" [20] (s/p).

Another aspect that does not bode well for inclusion is teacher training, specifically that of early childhood educators. Craveiro [34] and Esper et al. [24] demonstrate the non-existence of initial training systems that guarantee the development of the necessary competencies for future early childhood educators to meet all the demands related to their professional activity and, consequently, the needs of children. They suggest improvements such as an adequate balance between the components of knowledge; the development of theory and practice; the promotion of crucial competencies in enabling students to carry out their future professional functions; and the training and development of reflective and investigative practices.

## 3. Method

In the methodological framework, a mixed-nature study, descriptive and inferential, was chosen, using a questionnaire survey applied in mainland Portugal and the islands (Madeira and Azores archipelagos), involving a sample of 250 participants. The questionnaire (see Supplementary Materials) consisted of a limited number of closed-ended questions and one open-ended question, composed of three distinct parts, the first part referring to the research topic, research objectives, anonymity, and confidentiality. The second part included the sample characterization and data on the school situation of preschool children. Finally, the third part of the questionnaire included 24 closed-ended questions globally coded according to the Likert scale (24), and one open-ended question (1).

The study addressed the research question, "What is the impact of Decree-law no. 54/2018 on the inclusion of preschool children?"

Based on this research question, the following objectives were chosen: General objective: To analyze the perspectives of early childhood educators working in various teaching sectors on inclusion in EPE; Specific Objectives: 1. To understand the perspectives of early childhood education professionals on the application, principles, and definitions advocated in Decree-Law no. 54/2018; 2. To analyze the involvement of the multidisciplinary support team for inclusive education (EMAEI) resulting from the approval of the aforementioned Diploma; 3. To know the opinions of respondents regarding the training of early childhood

educators; 4. To assess the perception of early childhood educators regarding pedagogical work, collaboration, and support mobilized for EPE children.

The results were statistically analyzed to explore and deepen the problem already described. According to the defined research questions, we used the SPSS 24.0 Program for closed-ended questions. We applied the Kruskal–Wallis test to measure differences in opinions among early childhood educators from various teaching sectors. Simultaneously, in the interpretive analysis of data, we considered relative and absolute frequencies.

In the open question "Do you consider that the approval of Decree no. 54/2018, in Pre-School Education, aims to respond to the diversity of children's needs and has contributed to the involvement of all in the learning and inclusion process?", content analysis was employed (CA) [35] using the QDA Minor Program. It allowed us to organize the data based on an analogical sense of the chosen registration units, which grouped them into categories and subcategories [36]. The analysis resulted in the following category system:

1.  Category 1: Application of Decree-Law no. 54/2018
2.  Subcategory: Measures and strategies
3.  Category 2: Mission of preschool education
4.  Subcategory: Learning, projects, and pathways
5.  Category 3: Scopes of inclusion
6.  Subcategories: Diversity and involvement
7.  Category 4: Resource management
8.  Subcategories: Training and intervention of professional, physical, and financial resources

The fidelity of the results was obtained through the degree of internal consistency (Cronbach's alpha), with the questionnaire survey presenting a coefficient value of 0.863, falling under the "Good" category.

*Sample Characterization*

The sample consists of 250 respondents who completed the questionnaire, which was validated (pre-test) and placed *online* (via *Google docs*) to address difficulties arising from the COVID-19 pandemic. We found that 96.8% ($n = 242$) of the participants are female and 3.2% ($n = 8$) male, with ages ranging from 41 to 55 years, representing 59.6% ($n = 149$) of the sample. The majority of respondents reside in the "North" region of Portugal 29.2% ($n = 73$), followed by the "Metropolitan Region of Lisbon" (28.8%; $n = 72$), and the "Center" (25.6%; $n = 64$). As for academic qualifications, we observed that the incidence of participants with a "Bachelor's degree" was 69.2% ($n = 173$) and those with a "Master's degree" 27.2% ($n = 68$); 71.6% ($n = 179$) of respondents work in the "Public Education" sector and 58.8% ($n = 147$) have between 6 and 25 years of service experience (Table 1).

**Table 1.** Characterization of the sample.

| Characterization of the Sample | Absolute and Relative Frequencies ($n = 250$) |
| --- | --- |
| AGE | |
| Less than 25 years old | 1 (0.4%) |
| 26–40 years old | 49 (19.6%) |
| 41–55 years old | 149 (59.6%) |
| Older than 56 years | 51 (20.4%) |
| GENDER | |
| Female | 242 (96.8%) |
| Male | 8 (3.2%) |
| RESIDENCE AREA | |
| Alentejo | 16 (6.4%) |
| Algarve | 6 (2.4%) |
| Madeira Archipelago | 3 (1.2%) |
| Azores Archipelago | 16 (6.4%) |

**Table 1.** *Cont.*

| Characterization of the Sample | Absolute and Relative Frequencies (*n* = 250) |
|---|---|
| Center | 64 (25.6%) |
| North | 73 (29.2%) |
| Lisbon Metropolitan Region | 72 (28.8%) |
| ACADEMIC QUALIFICATIONS | |
| Bachelor's degree | 7 (2.8%) |
| Graduate degree | 173 (69.2%) |
| Master's degree | 68 (27.2%) |
| Doctorate degree | 2 (0.8%) |
| SECTOR OF EDUCATION | |
| Cooperative Education | 15 (6%) |
| Private Education | 17 (6.8%) |
| Public Education | 179 (71.6%) |
| IPSS (Institution of Social Solidarity) | 39 (15.6%) |
| YEARS OF SERVICE | |
| Less than 5 years | 14 (5.6%) |
| 6–25 years | 147 (58.8%) |
| More than 26 years | 89 (35.6%) |

In summary, the convenience sample from digital platforms dedicated to the topic of inclusion was the solution found to overcome the constraints and sanitary impositions resulting from the limitations and sanitary rules related to SARS-CoV-2 (during the data collection period) and the restrictions of the GDPR (still in force) regarding the contact information of early childhood educators in the continent and islands regarding the matter under consideration.

## 4. Results

Regarding the characterization of children's groups (Table 2), we found that 74% (*n* = 185) of the respondents indicated that their group is composed of children of different ages, with groups of heterogeneous ages prevailing. About 81.2% (*n* = 203) of respondents stated that they have children identified with MSAI; 55.2% (*n* = 138) are referred to with specific health needs (NSE); 72.4% (*n* = 181) benefit from universal measures, and 52.8% (*n* = 132) benefit from selective and/or additional measures. These data indicate that there is a very significant number of children who have been mobilized with MSAI. Therefore, we highlight that the majority of children benefit from universal measures. However, there is a considerable percentage of children with selective and/or additional measures, including those with NSE.

**Table 2.** Characterization of the groups of children.

| Characterization of the Groups of Children | Yes | No |
|---|---|---|
| Is your group composed of children within the same age range? | 65 (26%) | 185 (74%) |
| Are there children identified with measures to support learning and inclusion? | 203 (81.2%) | 47 (18.8%) |
| Are there children flagged as having specific health needs? | 138 (55.2%) | 112 (44.8%) |
| Are there children benefiting from universal measures? | 181 (72.4%) | 69 (27.6%) |
| In the group, are there referenced children with selective and/or additional measures? | 132 (52.8%) | 118 (47.2%) |

Next, we present a descriptive and inferential summary of the results obtained in the study, according to the research questions defined for this investigation. For research question 1: "*What are the perspectives of early childhood education professionals on the application, principles, and definitions of Decree-law no. 54/2018?*", we found that:

When asked the question "*Is the decree fully implemented in schools?*", 52.7% (*n* = 132) of respondents disagreed or strongly disagreed. When asked to give their opinion on whether

"*Educators show difficulties/doubts in applying the decree?*", 75.5% (*n* = 189) of participants agreed or strongly agreed. Regarding the item "*Is this decree bureaucratic?*", the majority (85.1%; *n* = 213) indicated that they agreed or strongly agreed. With regard to the question "*Is this Decree-Law functional, taking into account the pedagogical practices developed in preschool?*", 38.6% (*n* = 97) of participants disagreed or strongly disagreed, 36.9% (*n* = 92) agreed or strongly agreed, and 24.5% (*n* = 61) responded that they had no opinion. When asked if "*This decree brought something new to the inclusion of preschool children?*", 38% (*n* = 95) of respondents disagreed or strongly disagreed, 37.7% (*n* = 94) agreed or strongly agreed, and 24.4% (*n* = 61) were neutral on this issue.

Furthermore, regarding research question 1 and emphasizing the principles and definitions advocated for in the analyzed diploma, we found that when considering the question "*Are educators enlightened about the guiding principles of this Decree-Law?*", it was possible to verify that 51.5% (*n* = 129) of participants disagree or totally disagree. With regard to the assertion "*Do educators understand the nomenclature of the diploma?*", we found that 47.1% (*n* = 118) of respondents disagree or totally disagree. When asked if "*Educators are informed about the Multilevel Model?*", 59.7% (*n* = 149) stated that they disagree or totally disagree with this item. Also, with regard to the question "*Do preschool educators understand the principles of Universal Design for Learning?*", 45.3% (*n* = 113) of respondents stated that they disagree or totally disagree. Regarding the item "*Do you consider that selective and/or additional measures are not suitable for preschool education, and that all possibilities offered by a universal and preventive approach should be exhausted?*", 40.3% (*n* = 101) agree or totally agree and 39.8% (*n* = 99) disagree or totally disagree, with 19.9% (*n* = 50) having no formed opinion on this assertion.

When the Kruskal–Wallis test (non-parametric test used in the comparison of three or more independent samples) was applied, we found that regarding the question "*Is the diploma fully implemented in schools?*", the opinions among educators from different teaching sectors do not differ (*disagree, p-value = 0.078*). Similarly, regarding the question "*Do you consider that educators show difficulties/doubts in applying the diploma?*", we found that the opinions converge (*agree, p-value = 0.765*). Regarding the assertion "*Is this diploma bureaucratic?*", we found that opinions do not differ among preschool educators (*agree, p-value = 0.908*). As for the question "*Is this Decree-Law functional, taking into account the pedagogical practices developed in preschool?*", we verified that there are no significant differences in opinions among preschool educators (*no opinion, p-value = 0.901*). In the question "*Did this diploma bring something new to the inclusion of preschool children?*", we found that the opinions among educators from different teaching sectors also do not diverge (*no opinion, p-value = 0.057*).

Regarding the results of the Kruskal–Wallis test, we found that for the item "*Are educators knowledgeable about the guiding principles of this Decree-Law?*", opinions among educators from different teaching sectors do not differ (*disagree, p-value = 0.234*). Concerning the question "*Do educators understand the terminology of the diploma?*", all teachers disagree with this item, except for public school professionals who have no formed opinion, with differing opinions among teachers from various teaching sectors. For the question "*Are educators knowledgeable about the Multilevel Model?*", private school educators have a divergent opinion (completely disagree with the statement) from other educators. The remaining teachers disagree about the statement of the item. Regarding the question "*Do early childhood educators understand the principles of Universal Design for Learning?*", we found that opinions among educators do not differ (*no opinion, p-value = 0.225*). Concerning the item "*Do you believe that selective and/or additional measures are not suitable for preschool education, and that all possibilities offered by a universal and preventive approach should be exhausted?*", we found that opinions among educators from different teaching sectors converge (*no opinion, p-value = 0.155*).

Regarding research question 2 "What is the involvement of the Multidisciplinary Support Team for Inclusive Education (EMAEI) in the implementation of the current diploma?", we found that:

When asked *"Does EMAEI prioritize proximity to educators?"*, 38.4% (*n* = 96) agree or strongly agree, 36.7% (*n* = 92) of respondents disagree or strongly disagree, and 25% (*n* = 62) responded that they have no formed opinion. Regarding the question *"Does EMAEI regularly and effectively monitor measures to support children's learning and inclusion?"*, 41.3% (*n* = 103) disagree or strongly disagree and 36.1% (*n* = 90) agree or strongly agree. When asked *"Does EMAEI provide training to preschool educators?"*, 62.8% (*n* = 157) of respondents disagreed or strongly disagreed. Regarding the question *"Does EMAEI take into account the opinions of parents or guardians?"*, 51% (*n* = 128) agreed or strongly agreed. Finally, regarding the assertion *"Does EMAEI value the opinion of other variable elements besides parents?"*, 50.8% (*n* = 127) agreed or strongly agreed. In our open question, we found that "there are EMAEI teams that refuse to refer preschool children, only accepting severe and visible cases" and that "many educators, schools, and EMAEIs consider that Decree 54 does not apply to preschools". Therefore, support is provided by the ELI and lead educator: "EMAEI says that now they will only have support from ELI, and ELI says that they only give instructions (indirect support)".

When applying the Kruskal–Wallis test to the question *"Does the EMAEI prioritize proximity with educators?"*, we found that educators from different sectors of education do not have a formed opinion on this item, except for private sector professionals, who do not agree. Regarding the item *"Does the EMAEI regularly and effectively monitor measures to support children's learning and inclusion?"*, the opinions among educators do not differ (no opinion) (*p-value = 0.532*). Concerning the question *"Does the EMAEI provide training for early childhood educators?"*, we found that the opinions among educators converge (*disagree*) (*p-value = 0.694*). Regarding the question *"Does the EMAEI take into account the opinions of parents or guardians?"*, we found that the opinions among educators from different sectors of education do not differ (*no opinion*) (*p-value = 0.107*). Regarding the item *"Does the EMAEI value the opinions of other variable elements besides parents?"*, we found that the opinions among educators also do not differ (*no opinion*) (*p-value = 0.203*).

Regarding research question 3 "What are the respondents' opinions on the training of early childhood educators?", we found that:

When asked "Do university curricula provide adequate scientific knowledge about Inclusive Education?", 52.4% (*n* = 131) of respondents disagree or strongly disagree. In response to the question *"Do educators feel the need for continuous training?"*, we found that 95.5% (*n* = 239) agree or strongly agree. Regarding the question "Do you believe that the lack of training for educators is a hindrance to the implementation of the principles of the law?", 77% (*n* = 193) agree or strongly agree.

In the open question, we found that the lack of training is a barrier in the inclusion process, with participants expressing that "without basic training for Educators of Infancy and more human resources, it will be difficult to fulfill what is on paper".

Based on the Kruskal–Wallis test, we found that for the question *"Do university curricula ensure proper scientific knowledge about Inclusive Education?"*, the opinions among educators from different teaching sectors do not differ (*disagree*) (*p-value = 0.254*). Regarding the question *"Do educators feel the need for continuous training?"*, educators from private and public teaching sectors agree with the statement, while professionals from cooperative education and social solidarity institutions (IPSS) strongly agree with it. For the question *"Do you consider the lack of training for educators to be a constraint in implementing the principles of the diploma?"*, we found that the opinions among educators from different teaching sectors do not differ (*agree*) (*p-value = 0.694*).

For research question 4, which seeks to analyze "What perceptions do early childhood educators have regarding pedagogical and collaborative work and the supports mobilized for preschool education children?", we found that:

When asked the question "Do preschool educators plan respecting the learning rhythms of children with greater difficulties?", 72% (*n* = 180) agree or strongly agree. Regarding whether "Preschool educators carry out pedagogical differentiation in the classroom context", 74.8% (*n* = 187) agree or strongly agree. Facing the question "Do you

consider collaborative work among different technicians and specialists to be an enrichment for preschool educators?", 86.8% ($n = 217$) agree or strongly agree. When asked "Do you consider the supports provided to children with greater difficulties to be adequate?", 58.4% ($n = 146$) disagree or strongly disagree. Regarding the question "Are the supports provided by the school sufficient to overcome the difficulties of children in preschool education?", 79.2% ($n = 198$) disagree or strongly disagree. Regarding the item "Are there children who were left without support after the approval of Decree-Law no. 54/2018?", 47.2% ($n = 118$) agree or strongly agree. However, we obtained a significant number of participants who do not have a formed opinion on this matter, namely 31.2% ($n = 78$).

According to the Kruskal–Wallis test, we verified that regarding the question "Do early childhood educators plan while respecting the learning rhythms of children with more difficulties?", cooperative education, public education, and IPSS holders agree with the statement, while private education professionals claim not to have a formed opinion on this item. Regarding the question "Do early childhood educators carry out pedagogical differentiation in the classroom context?", cooperative education, public education, and IPSS holders agree, while private education professionals claimed not to have a formed opinion on this question. In relation to the assertion "Do you consider collaborative work among different technicians and specialists an enrichment for early childhood educators?", the opinions among early childhood educators from different educational sectors do not differ (agree) ($p$-value = 0.845). When analyzing the item "Do you consider the supports provided to children with more difficulties adequate?", we ascertained that the opinions among teachers do not differ (disagree) ($p$-value = 0.979). When asked whether "The supports provided by the school are sufficient to overcome the difficulties of children in preschool education", we ascertained that the opinions among teachers do not differ (disagree) ($p$-value = 0.489). Finally, regarding the question "Have there been children who have been left without support after the approval of Decree-Law No. 54/2018?", we also found that the opinions among early childhood educators from different educational sectors do not differ (no opinion formed) ($p$-value = 0.520).

The analysis revealed that the first category and its corresponding subcategory had the highest number of recording units, totaling 45, which accounted for 35.7% of the total. Conversely, the fourth category, specifically its second subcategory, had the lowest number of recording units, with only 7, representing 5.6% of the total. Esteves [35] suggests that a category with a higher number of recording units holds greater importance compared to others.

Based on these findings, the study determined the following:

Regarding the application of Decree-Law no. 54/2018, the measures and strategies had both positive aspects, such as generating new opportunities and providing inclusive education for all, and negative aspects, including the devaluation of students in certain measures, non-application of measures in preschool and private education, inequality in implementing measures and strategies across schools, mismatch between theory and practice, bureaucratic application process, and lack of training for teachers. Respondents emphasized the negative aspects more strongly.

In terms of the mission of preschool education, which includes learning, projects, and pathways, there were potentialities such as flexible principles, commitment to inclusive work, promotion of informal learning, and response to children's specificities. However, limitations were also identified, such as the non-application of selective measures and additional services, ineffective signage, and devaluation of pedagogical work due to lack of support.

The analysis also highlighted two areas of inclusion: diversity and involvement. In terms of diversity, there were positive aspects such as recognizing the abilities of each student, responding to the diversities of children, and setting specific goals for each student. However, limitations were observed, including limited support for disadvantaged and unprotected children, and the failure to achieve objectives of pedagogical differentiation.

Regarding involvement, the school was recognized as having the function of involving all stakeholders and a commitment to training for enhancing inclusion, as well as focusing on the specific needs of children. However, some limitations were noted, such as perceiving inclusion as a negative aspect of involvement, limited support from technical teams, and the need to adapt measures to different children.

Finally, the analysis highlighted the management of human resources within two dimensions:

1. The training and intervention of professionals had positive aspects such as the need for continuous training and showcasing pedagogical work, but also negative aspects including staff without qualifications/competences, reduction in professionals with adequate training, the team as an influencer of intervention, devaluation of the childhood educator career, scarcity of professionals in relation to needs, the need for EMAI teams to review their practices, universal measures not aligned with reality, lack of support for children, families, and schools, lack of knowledge of difficulties, low demand for training due to age and motivation factors, and associated formality to intervention.

2. Physical and financial resources played a role, with spatial conditions influencing inclusion and the limited availability of resources affecting the provision quality service and addressing specific needs, as well as selective support not being universally accessible.

In the open-ended question, we were able to identify a high level of dissatisfaction among preschool teachers regarding the implementation and knowledge of the law, as they expressed that "*there is still a lot of ignorance about the DL*"; "*most of the school groups do not put into practice what the law provides for and this is the real obstacle*". *In the opinion of the respondents, it is a bureaucratic diploma, registering that* "*the decree at this moment functions as mere bureaucracy*".

## 5. Discussion

In response to research question 1, "What are the perspectives of early childhood education professionals regarding the application, principles and definitions of Decree-Law No. 54/2018?", the following results were obtained:

According to the kindergarten teachers who took part in this study, there is a consensus that the policy regulating inclusive education in Portugal is not being fully implemented in Portuguese schools.

Previous studies, namely Monteiro et al., argue that "there is still a lack of knowledge about the DL by the majority of school professionals" [37] (p. 76), with the participants in the present research expressing insecurity and confusion: "I am completely confused"; "I don't feel secure", highlighting the lack of preparation of educators and teachers regarding the analyzed diploma due to a marked "lack of knowledge of measures and strategies" (CA), as expected. As for the existing doubts, "those that nobody clears up, almost afraid to ask the authorities because they are not consistent in their answers, it depends on who you ask" [37] (p. 83), with bureaucratization presenting itself as a hindrance in the inclusion process. Thus, we found that "there is bureaucracy in the application process" (CA). The results of this research corroborated the data we obtained and analyzed.

Additionally, the results suggest that educators lack proper information and clarity regarding the principles outlined in this Decree-Law, as well as the multilevel model. Corroborating with the data, Bonança et al. [38] argue that it is of utmost urgency to empower the entire educational community on the principles and terminology of the diploma, raising the quality of educational practices, supported by the universal design for learning (UDL). Regarding the multilevel model, Colôa argues that in "the legislation now under consideration, the multilevel model presents itself as a hybrid conceptualized from a perspective of organizing educational measures that are configured as circumscribed and prescriptive responses to the expected diversity of students that make up 21st-century

schools" [39] (p. 34), implying that educators and teachers are not properly informed about the multilevel model that defines MSAIs, as confirmed by the data from our study.

Although in smaller numbers, it was observed that some educators have difficulty understanding the principles of the universal design for learning, which consequently affects the academic success of students requiring additional school support. According to the respondents' responses on the question "Do you consider that selective and/or additional measures are not adequate for pre-school education, and that all the possibilities that a universal and preventive approach should be exhausted should be used?", we found no conclusive position.

The inferential analysis of the data indicates that educators from different educational sectors hold similar opinions regarding all items within this group. However, opinions differ on the question "Are educators clear about the Multilevel Model?". Contrary to other educators, those from the private education sector hold a divergent opinion, completely disagreeing with the statement. The remaining professors also disagree with the statement.

We found that, at the time of this questionnaire survey, preschool educators were not adequately enlightened about the guiding principles of the Decree-Law, had difficulties in understanding the nomenclature, doubts about the multilevel model and what is advocated by the universal design for learning, data that reflect the need for a transition period between DL no. 54/2018 and DL no. 3/2008 [38]. Based on the findings of this research, it is evident that there is a need for a transition period between Decree-Law no. 54/2018 and Decree-Law no. 3/2008 [38].

Regarding research question 2, "What is the involvement of the Multidisciplinary Team for the Support of Inclusive Education (EMAEI), in view of the implementation of the diploma in force?", as "EMAEI monitors, on a regular and effective basis, measures to support the learning and inclusion of children", we cannot make any conclusions about these items. Furthermore, during the research, it was observed that the EMAEI does not provide training for kindergarten teachers, contributing to the identified "lack of training for teachers" (AC). In addition, there is a "lack of articulation between EMAEI technicians" (AC), indicating a need for better coordination and collaboration among these professionals. Moreover, the EMAEI tends to prioritize the opinions of parents or guardians.

The results of the study conducted by the FNE [32] differ from those of our research regarding the evaluation of EMAEI's parent or guardian participation in the inclusion process.

The inferential analysis of the results suggests that, overall, the opinions of other educators across different teaching sectors do not differ significantly, except for the statement regarding whether the EMAEI prioritizes proximity to educators.

With regard to research question 3, "What are the opinions of respondents about the training of kindergarten teachers?", it was observed that the curricular plans of universities do not adequately provide the necessary scientific knowledge about inclusive education. Almost all respondents emphasized the importance of educators engaging in continuous training, as the lack of such training was identified as a hindrance in effectively implementing the diploma. Consequently, it can be concluded that there is a recognized "need for continuous training to improve practice" (AC) in the field of inclusive education.

The results of the Kruskal–Wallis test reveal that there is a significant difference among educators from various education sectors in their opinions on the question "Do educators feel the need to carry out continuous training?" However, they hold a similar position on the remaining issues addressed in the study.

The data from this study on the initial and ongoing training of educators find resonance in the extensive review of the specialized literature [39–42].

For research question 4, which seeks to analyze "What perceptions do kindergarten teachers have in relation to pedagogical, collaborative work and the support mobilized for children in preschool education?", the research findings indicate that a significant number of kindergarten teachers engage in planning activities that take into account the learning rhythms of children with more difficulties; furthermore, these teachers also

practice pedagogical differentiation within the classroom context. In this context, it becomes evident that there is a clear "recognition of the abilities of each student" (AC) and a strong commitment to "responding to the diversity of children" (AC). However, it is important to note that UNESCO has highlighted the potential consequences of inadequate preparation for inclusive education, attributing it to gaps in pedagogical knowledge [27] (p. 20).

Additionally, the data from the study indicate that kindergarten teachers who voluntarily participated in this investigation strongly believe that collaborative work with various technicians and specialists enriches the inclusion process. This emphasizes the importance of recognizing that "the school has the function of involving all stakeholders" (AC). It is crucial to promote a growing commitment to training and professional development to enhance inclusive practices and foster an inclusive learning environment.

Regarding the support provided to children with difficulties, the findings indicate that the support offered is not always suitable or sufficient to address the gaps that these children have. Some educators even reported instances where children were left without any kind of support following the approval of Decree-Law no. 54/2018. This is highlighted by the observation that there is a "lack of support for children, families, and schools" (AC).

Furthermore, regarding the question "Do kindergarten teachers plan, respecting the learning rhythms of children with more difficulties?", it was found that there are differences in opinions between educators working in cooperative and public education compared to those working in the IPSS (Instituição Particular de Solidariedade Social).

Similarly, concerning the question "Do early childhood educators perform pedagogical differentiation in the classroom context?", opinions vary among educators from cooperative education, IPSS, public education, and private education.

However, for other issues addressed in the study, educators hold similar opinions across different sectors of education.

Furthermore, in light of the collected data for the open-ended question, we found that regarding the terminology, "*the novelty will be the terminology of more specific supports, but INCLUSION cannot be decreed. It has to 'happen'*"., the situation becomes more complex when participants perceive compromising data that reflect the unstable situation experienced in preschools, stating that "*no colleague with a degree or master's degree has any idea what is expected, they sign the documents presented to them blindly, few do not know how to fill them out and assume commitments they are unaware of*". This, therefore, is implicit of educators' lack of knowledge about the principles and definitions of the analyzed diploma.

In the open-ended question, there are significant data confirming the lack of support for preschool children: "Many don't even get support. Because special education teachers are insufficient for so many children in need. More and more"; "The support teams for children with special needs still can't respond to the real number of children who unfortunately need it"; "the approval of the decree has meant that preschool children have no type of support, except for that provided by the teacher"; "there are preschool children with pathologies that require additional or selective measures already at an early age". As for the extension of more restrictive measures to preschool children, respondents believe that "preschool children cannot have selective, let alone additional, measures, so the children are deeply prejudiced. I just regret it!" Regarding collaborative work, they make the following accusation: "as for EMAIE, they don't even bother to hold meetings with all fixed members, the coordinator presents what she can offer and it's take it or leave it. So, I regret the opportunism that this decree made possible".

## 6. Conclusions

Based on the data collected, we can conclude that there is still a long way to go regarding the inclusion of children in early childhood education. First, we list the main results that require intervention and future improvements: 1. High dissatisfaction among early childhood educators regarding the implementation of DL no. 54/2018, five years after its approval; 2. Incomplete implementation of the studied DL, with doubts and difficulties persisting regarding its application; 3. The high bureaucracy foreseen in this DL

is a disadvantage in the inclusion process; 4. Educators do not feel adequately informed about the nomenclature and guiding principles of this DL, as well as the principles stated in the universal design for learning; 5. The support provided for children in early childhood education is insufficient to meet their needs; 6. The EMAEI should monitor children with MSAI more regularly and efficiently; 7. There is a need to improve and reinforce the initial training of educators in the area of inclusive education; and 8. There is a need to strengthen the ongoing training of professionals. These last two (7 and 8) are significant limitations in implementing the principles advocated in the diploma.

We also present the main results for which there are no conclusive and significant data that raise future intervention and concern: 1. The DL seems to be relatively functional for the pedagogical practices developed in preschool; 2. The DL brought something new to the inclusion of EPE children; 3. Selective and/or additional measures seem to be suitable for EPE; 4. There are no substantial differences of opinions regarding the understanding of the nomenclature of the diploma and the degree of clarification of the multilevel model, despite public sector teachers having a divergent opinion; and 5. The EMAEI prioritizes proximity to educators.

Finally, we highlight not only that the EMAEI significantly welcomes the opinions of parents or guardians, but also the overall quality of the pedagogical work developed by educators of infancy, both in planning activities and in pedagogical differentiation for children who need more support and assistance in their learning development. Collaborative work between technicians and specialists is seen as an enrichment of the inclusive process as a whole.

We strongly believe that there is an urgent need to carry out a comparative study in the near future on the legislation produced relating special education and inclusion in the European context, along with similar legislation in other continents, based on the scientific literature, legislation, and international reports produced over the last decade. However, a study with these terms and purposes, with a more global vision and perception of the topic on the European and international spectrum, may come up against some of the limitations we face in the present study. This is because the national characterization and specificity, even if framed in the general European and international legal and normative framework, tends to limit any broader extrapolation of the results at a global level. This could be considered a clear limitation, since a global study in various countries with specific and adapted legislation could result in a list of indicators and metrics of legislation produced within a certain set of countries or states (e.g., EU or USA). On the other hand, if there is, in fact, a certain limitation within the generalization or non-intention of generalization of the study data, given its peculiarity or specificity, it is possible to generalize the results of studies in similar contexts with a certain degree of confidence, where this generalization is established through the analysis of similar characteristics between particular cases [43].

**Supplementary Materials:** The following supporting information can be downloaded at: https://www.mdpi.com/article/10.3390/educsci13070737/s1, Questionnaire.

**Author Contributions:** R.S.B.: Conceptualization, methodology, data curation, resources, writing—original draft, investigation, writing—review and editing; P.C.F.B.: Methodology, formal analysis; L.L.: Supervision, validation, visualization, writing—review and editing.; E.M.G.M.: Supervision, validation, visualization, writing—review and editing. All authors have read and agreed to the published version of the manuscript.

**Funding:** This research received no external funding.

**Institutional Review Board Statement:** Not applicable.

**Informed Consent Statement:** Informed consent and voluntary participation all subjects were obtained, with anonymity.

**Data Availability Statement:** All data generated or analyzed during this study are included in this published article.

**Conflicts of Interest:** The authors declare no conflict of interest.

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
