# Peer review of "Decree-Law 54/2018: Perspectives of Early Childhood Educators on Inclusion in Preschool Education in Portugal"

_education, doi:10.3390/educsci13070737_

Round 1
Reviewer 1 Report
In relation to the article entitled "Decree-law 54/2018: perspectives of early childhood educators on inclusion in preschool education", indicate that:
The authors have made some improvements in the manuscript, among which is the inclusion of a paragraph within the theoretical part that expands the information provided on the research topic at the national level, which should be valued. They have also included some reflections (although they are brief) on the needs derived from the conclusions of this study.
On the other hand, I think it is important that the authors improve the manuscript in the following aspects:
- The authors bring together results and discussion, sections that should be shown separately.
- The results could be improved, since only an analysis is made according to items or questions raised. Some more advanced analysis technique could be applied to give depth to the findings obtained and to the contrast of hypotheses.
- Although the discussion is linked to the results, it is quite poor. It is recommended that it be expanded in a different section, analyzing the scientific literature in depth and contrasting the findings obtained with those of other authors. The given limitations should be deepened, and above all, the future perspectives should be developed more solidly.
Author Response
Thank you for reading our manuscript and offering constructive feedback. We appreciate your suggestions and have made major revisions. Below, please find the reviewer’s comments with our responses in italics in red.
REVIEWER 1: The authors have made some improvements in the manuscript, among which is the inclusion of a paragraph within the theoretical part that expands the information provided on the research topic at the national level, which should be valued. They have also included some reflections (although they are brief) on the needs derived from the conclusions of this study. On the other hand, I think it is important that the authors improve the manuscript in the following aspects:
- The authors bring together results and discussion, sections that should be shown separately.
The sections (points) related to the results and discussion, as well as the further discussion, were separated in two moments. Namely: 1. in the last three paragraphs of section 4. "Results", where the most significant results obtained from the triangulation of the data generated from the methodological tools (e.g. categories and subcategories) and dimensions used in this research are analysed, with particular emphasis on specific indicators that will be discussed in more detail later; 2. In paragraphs 1, 2, 3, 4, 5, 6, 7, 9, 10, 11, 13, 14, 15 and 16 of item 5 "Discussion" we detail, part by part, question by question, the discussion of the results obtained from the methodological tools used, and the possible impact on future conclusions arising therefrom.
- The results could be improved, since only an analysis is made according to items or questions raised. Some more advanced analysis technique could be applied to give depth to the findings obtained and to the contrast of hypotheses.
In this sense, in paragraph 5 of section 3, "Method", information on content analysis was added and the respective categories and subcategories used in the study were indicated. It also mentions the fidelity of the results obtained through the degree of internal consistency. The information about the open question (of qualitative nature) regarding the content analysis has been added, which is developed, after this introduction, in two different moments. In the 'last paragraph of section 4, "Results", immediately before section 5, "Discussion"; 2. in the last two paragraphs of section 5, "Discussion", immediately before section 6, "Conlusions", where this issue is explained in detail.
- Although the discussion is linked to the results, it is quite poor. It is recommended that it be expanded in a different section, analyzing the scientific literature in depth and contrasting the findings obtained with those of other authors. The given limitations should be deepened, and above all, the future perspectives should be developed more solidly.
A different (and separate) section was added in order to meet the requirements of reviewer 1. This can be seen in the subdivision and deepening of the discussion in items n. 4 "Results" and n. 5 "Discussion". We kept (even though added) item n. 6 "Conclusions". The general conclusions of the research were also added and deepened, along with the explanation regarding the limitations of the study (last paragraph of the article).
Important note: The article has been revised and corrected for minor typos or terms that we found in response to the request of reviewer 2, including in the text added in response to the requests of reviewer 1. Many paragraphs were also (totally or partially) rewritten, of which we highlight the following: Page 6 (Paragraphs: 3,4); Page 10 (Paragraphs: 3,4, and 5); Page 11 (Paragraphs: 1, 2, 3, 5, 6, 7 and 8); Page 12 (Paragraphs: 2, 3, 4, 5, 6 and 7).
We are grateful for all suggestions for improvement of the manuscript.

Reviewer 2 Report
The article is focused on a relevant and important field of the inclusive education - however, I would expect a specification of national focus even in a title/headline (all the more so when there is a question on particular national degree in Portugal...).
I appreciate the detailed overview of the evolution as well as the current situation in a legal setting of preschool and primary school education of pupils with special educational needs in Portugal. Moreover, there is also a short paragraph considering the international and European educational context. On the other hand, it is not clear for me why there is the COVID-19 pandemic mentioned so often in connection with such a wide and exceeding topic, as it did not brought so many changes into the concept of inclsuive education itself. However, the descriptive outline of inclusive education conditions and parameters was welcome and it corresponds to the research topic as well.
The critical discussion on some challenges of the implementation of inclusive in education was appropritate for this study, although in some points it was too much skeptic on this kind of educational approach.
We can find clearly stated research questions and objectives in the research report, as well as an introduction of research methodology and design: the quantitative part used a complex questionnaire and the data was analyzed by the advanced statistic tools and methods, while the qualitative part was represented just in one open question at the questionnaire, and we just read that the content analysis was used in this part, without many details on qualitative results at all.
It seems that some reserch questions relied on a subjective opinion of respondents at all (i.e. Do you think the decree is implemented...? - p.7-8, etc.). However, each part is compared to the results of some other relevant published research reports, which was very interesting.
As for the ethic rules applied in research process, we can find a few words on this issue in the article.
In a section of References, we can see that most resources were choosed in Portugeese literature, while only a few items are in English and considering the international field as well...
Finally, I would recommend the proofreading of the document, so that the quality of English language is improved (especially the professional terminology used in an international context today...).
Author Response
Thank you for reading our manuscript and offering constructive feedback. We appreciate your suggestions and have made major revisions. Below, please find the reviewer’s comments with our responses in italics in red.
REVIEWER 2:
The article is focused on a relevant and important field of the inclusive education - however, I would expect a specification of national focus even in a title/headline (all the more so when there is a question on particular national degree in Portugal...).
We have changed the title in order to respond to the request of Reviewer 2: “Decree-law 54/2018: perspectives of early childhood educators on inclusion in preschool education in Portugal”
I appreciate the detailed overview of the evolution as well as the current situation in a legal setting of preschool and primary school education of pupils with special educational needs in Portugal. Moreover, there is also a short paragraph considering the international and European educational context. On the other hand, it is not clear for me why there is the COVID-19 pandemic mentioned so often in connection with such a wide and exceeding topic, as it did not brought so many changes into the concept of inclsuive education itself. However, the descriptive outline of inclusive education conditions and parameters was welcome and it corresponds to the research topic as well. The critical discussion on some challenges of the implementation of inclusive in education was appropritate for this study, although in some points it was too much skeptic on this kind of educational approach.
The discussion was deepened due to the separation in two blocks (Results and Discussion) by indication of reviewer 1.
We can find clearly stated research questions and objectives in the research report, as well as an introduction of research methodology and design: the quantitative part used a complex questionnaire and the data was analyzed by the advanced statistic tools and methods, while the qualitative part was represented just in one open question at the questionnaire, and we just read that the content analysis was used in this part, without many details on qualitative results at all.
The information on the open question (of qualitative matrix) related to the content analysis was added, which is developed, after this introduction, in two different moments. In the 'last paragraph of section 4 "Results", immediately before section 5 "Discussion"; 2. in the last two paragraphs of section 5 "Discussion", immediately before section 6 "Conlusions", where this issue is explained in detail.
It seems that some reserch questions relied on a subjective opinion of respondents at all (i.e. Do you think the decree is implemented...? - p.7-8, etc.). However, each part is compared to the results of some other relevant published research reports, which was very interesting.
Thank you for your comment and analysis.
As for the ethic rules applied in research process, we can find a few words on this issue in the article.
Thank you for your comment and analysis.
In a section of References, we can see that most resources were choosed in Portugeese literature, while only a few items are in English and considering the international field as well...
The bibliographical references are mostly national, since this is a study on a specific portuguese legislation. The remaining bibliography has a direct focus on European and international studies and regulations, in order to contextualise the research at a supranational level.
Finally, I would recommend the proofreading of the document, so that the quality of English language is improved (especially the professional terminology used in an international context today...).
Important note: The article has been revised and corrected for minor typos or terms that we found in response to the request of reviewer 2, including in the text added in response to the requests of reviewer 1. Many paragraphs were also (totally or partially) rewritten, of which we highlight the following: Page 6 (Paragraphs: 3,4); Page 10 (Paragraphs: 3,4, and 5); Page 11 (Paragraphs: 1, 2, 3, 5, 6, 7 and 8); Page 12 (Paragraphs: 2, 3, 4, 5, 6 and 7).
We are grateful for all suggestions for improvement of the manuscript.

Round 2
Reviewer 2 Report
I appreciate the changes made after the recommendations from the first review - the quality of the article is better now.